# Arginine Expedites Erastin-Induced Ferroptosis through Fumarate

**DOI:** 10.3390/ijms241914595

**Published:** 2023-09-27

**Authors:** Xinxin Guo, Yubo Guo, Jiahuan Li, Qian Liu, Hao Wu

**Affiliations:** 1State Key Laboratory of Agricultural Microbiology, College of Veterinary Medicine, Huazhong Agricultural University, Wuhan 430070, China; guoxinxin@webmail.hzau.edu.cn (X.G.); guoyubo@webmail.hzau.edu.cn (Y.G.); jiahuanli@webmail.hzau.edu.cn (J.L.); lqian0792@webmail.hzau.edu (Q.L.); 2Hubei Hongshan Laboratory, Wuhan 430070, China

**Keywords:** ferroptosis, arginine, fumarate, argininosuccinate lyase, urea cycle

## Abstract

Ferroptosis is a newly characterized form of programmed cell death. The fundamental biochemical feature of ferroptosis is the lethal accumulation of iron-catalyzed lipid peroxidation. It has gradually been recognized that ferroptosis is implicated in the pathogenesis of a variety of human diseases. Increasing evidence has shed light on ferroptosis regulation by amino acid metabolism. Herein, we report that arginine deprivation potently inhibits erastin-induced ferroptosis, but not RSL3-induced ferroptosis, in several types of mammalian cells. Arginine presence reduces the intracellular glutathione (GSH) level by sustaining the biosynthesis of fumarate, which functions as a reactive α,β-unsaturated electrophilic metabolite and covalently binds to GSH to generate succinicGSH. siRNA-mediated knockdown of argininosuccinate lyase, the critical urea cycle enzyme directly catalyzing the biosynthesis of fumarate, significantly decreases cellular fumarate and thus relieves erastin-induced ferroptosis in the presence of arginine. Furthermore, fumarate is decreased during erastin exposure, suggesting that a protective mechanism exists to decelerate GSH depletion in response to pro-ferroptotic insult. Collectively, this study reveals the ferroptosis regulation by the arginine metabolism and expands the biochemical functionalities of arginine.

## 1. Introduction

Ferroptosis was originally defined in 2012 by Brent R. Stockwell and colleagues [1]. This novel form of programmed cell death is executed by the lethal accumulation of lipid peroxidation catalyzed by the intracellular bioactive iron, although the mechanism by which lipid peroxidation exactly triggers ferroptotic cell death is only partially understood [2]. GPX4 (glutathione peroxidase 4) and the co-factor GSH constitute the major cellular antioxidant system that counteracts ferroptotic cell death by detoxifying lipid peroxyl radicals and maintaining redox homeostasis [3]. Erastin, the first characterized ferroptosis inducer, targets the System Xc- and restrains cystine uptake, therefore reducing GSH biosynthesis, decreasing GPX4 activity, and eventually triggering ferroptosis [4]. Furthermore, RSL3, another typical ferroptosis inducer, directly binds to and suppresses GPX4, leading to the overwhelming accumulation of lipid peroxidation and initiating ferroptosis [5]. In addition, the FSP1 (ferroptosis suppressor protein 1)/CoQ10 (coenzyme Q10) axis [6], the DHODH (dihydroorotate dehydrogenase)/CoQ axis [7] as well as the GCH1 (guanosine triphosphate cyclohydrolase 1)/BH4 (tetrahydrobiopterin) axis [8] could catalyze the biosynthesis of metabolites with radical-trapping activity, and thus compensate the GPX4/GSH axis to reduce phospholipid hydroperoxides and suppress ferroptosis. Disruption of these endogenous antioxidative pathways could facilitate lipid peroxidation and initiate ferroptosis [1]. In a reverse reaction, LOXs (lipoxygenases) and POR (cytochrome p450 oxidoreductase) were nominated as the major enzymes that catalyze the generation of lipid peroxidation [9]. Moreover, the intracellular bioactive iron facilitates lipid peroxidation through mediating the Fenton reaction and/or sustaining the enzymatic activities of LOXs and POR [10].

Increasing evidence has emerged that several metabolic pathways could sense the intracellular and extracellular clues, and therefore dictate the ferroptosis vulnerability [11]. Among them, iron metabolism, lipid metabolism, and amino acid metabolism are the three important pathways. Specifically, the sulfur-containing proteinogenic amino acid, cysteine, is the key precursor and rate-limiting substrate of GSH [12]. Cysteine metabolism, especially cysteine uptake (cells absorb cysteine in its oxidized form, cystine, through the cystine-glutamate antiporter System Xc-, which is the exact target of erastin) as well as de novo cysteine synthesis (mainly referring to the transsulfuration pathway, which channels methionine to cysteine and other metabolites), determines the intracellular cysteine for the biosynthesis of cysteine-based metabolites including GSH, CoQ, and other ferroptosis-suppressive molecules [13]. Furthermore, glutamine metabolism is involved in the generation of reactive oxygen species (ROS) in mitochondria under pro-ferroptotic conditions [14]. Inhibition of glutamine uptake or glutamine-fueled glutaminolysis sharply disrupts erastin-induced ferroptosis [15]. In addition, the essential amino acid tryptophan has recently been identified as a key amino acid that counteracts ferroptotic cell death [16]. Specifically, tryptophan metabolites including indole-3-pyruvate [17], serotonin [18], 3-hydroxyanthranilic acid [18], and kynurenine [19] could directly trap free radicals and/or indirectly scavenge ROS through activating the endogenous antioxidative systems.

As a non- or semi-essential amino acid, arginine plays an important role in cellular metabolism and cell survival [20]. Arginine is the metabolic precursor of nitric oxide, urea, creatine, polyamines, proline, glutamate, guanabutamine, and hyperarginine, which are bioactive metabolites in distinct processes [21]. Moreover, it has been widely recognized that arginine can sustain mTOR (mammalian target of rapamycin) activation in the lysosomal surface, although the underlying mechanism is far from being thoroughly understood [22]. Arginine depletion by arginine-metabolizing enzymes including arginine deiminase and arginase induces cell death in cancer cells, especially in cancer cells with a lower expression of ASS1 (argininosuccinate synthase 1, which catalyzes citrulline and aspartate to synthesis argininosuccinate in the urea cycle) [23]. In 2021, Scott J. Dixon and his colleagues utilized the scalable time-lapse analysis of the cell death kinetics approach to quantify the potential ferroptosis regulation by different amino acids [24]. Arginine was nominated as an indispensable amino acid in ferroptosis execution in HT1080 cells in response to erastin2 (an erastin analog) or cystine withdrawal. Notably, arginine deprivation counteracts ferroptosis, which is independent of the mTOR and GCN2 (general control nonderepressible 2)-ATF4 (activating transcription factor 4) signaling pathways. However, the detailed mechanism is unclear. In this study, we present evidence that shows that arginine is essential for erastin-induced ferroptosis, but not RSL3-induced ferroptosis, in MEF and HT1080 cells. Mechanistically, arginine presence facilitates ferroptosis through decreasing intracellular GSH. In the urea cycle, arginine sustains the biosynthesis of fumarate, which functions as a reactive α,β-unsaturated electrophilic metabolite and covalently binds to GSH to generate succinicGSH. siRNA-mediated knockdown of *Asl* (argininosuccinate lyase), the urea cycle enzyme directly catalyzing fumarate biosynthesis, significantly reduces cellular fumarate and thus relieves erastin-induced ferroptosis in the presence of arginine. Notably, erastin exposure in MEF cells results in a decrease in intracellular fumarate. Collectively, this study further suggests ferroptosis regulation by arginine metabolism and expands the biochemical functionalities of arginine.

## 2. Results

### 2.1. Arginine Deprivation Inhibits Erastin-Induced Ferroptosis, but Not RSL3-Induced Ferroptosis

It has been reported that arginine deprivation mitigates ferroptosis in several types of mammalian cells [24]. However, the underlying mechanism is unclear. To determine the mechanism of arginine manipulating ferroptosis, MEF and HT1080 cells were treated with typical ferroptosis inducers (erastin and RSL3) in the complete or arginine-depleted medium. Propidium iodide (PI) staining showed that erastin and RSL3 exposure triggered notable cell death in the complete medium, while this cell death was almost completely abolished by the ferroptosis inhibitor Ferrostatin-1, suggesting that this cell death was indeed ferroptosis (Figure 1A,C). This was further confirmed by the cell viability assay. Erastin challenge led to a significant reduction in cell viability in the complete medium, and this could be largely suppressed by Ferrostatin-1 (Figure 1B). In this context, we found that arginine deprivation significantly inhibited erastin-induced ferroptosis (Figure 1A,B), but failed to inhibit RSL3-induced ferroptosis in the MEF and HT1080 cells (Figure 1C), which was consistent with the previously reported results [24]. In addition, erastin exposure triggered ferroptosis in a dose- and time-dependent manner in the MEF and HT1080 cells. We observed that arginine deprivation could suppress ferroptosis induced by the relative higher dose of erastin and by the relative longer time of erastin exposure (Appendix A). By compensating arginine into the arginine-depleted medium, we found that arginine manipulated ferroptosis sensitivity in a dose-dependent manner, as shown by the PI staining and cell viability analysis (Figure 1D,E).

Accumulation of lipid peroxidation and ROS boost are two typical hallmarks of ferroptosis [25]. By using C11-BODIPY 581/591, the fluorescent ratio-probe of lipid peroxidation, we found that arginine deprivation significantly reduced lipid peroxidation, which was elevated by erastin challenge in MEF and HT1080 cells (Figure 1F,G). In addition, total ROS measurement by H2DCFDA staining further suggested that erastin exposure elevated the total ROS, which was largely suppressed by arginine depletion (Figure 1H). GPX4 is the major anti-ferroptotic protein and could reduce peroxidized phospholipids with the assistance of its co-factor GSH. Erastin exposure decreased GPX4 protein. It was interesting that arginine depletion gave rise to negligible change in GPX4 expression, both under the steady state and under the pro-ferroptotic state (Figure 1I), implying that arginine manipulated ferroptosis in a GPX4-independent manner.

In summary, these data suggest that arginine deprivation inhibits erastin-induced ferroptosis, but not RSL3-induced ferroptosis, in mammalian cells.

### 2.2. Arginine Promotes Erastin-Induced Ferroptosis by Reducing GSH Levels

Erastin initiates ferroptosis by restraining cystine uptake through inhibiting System Xc-, leading to reduced GSH synthesis and the following accumulation of lipid peroxidation [4]. Erastin triggered a fast GSH exhaustion in the complete medium, while arginine deprivation elevated the GSH levels under both the steady state and erastin-treated condition (Figure 2A). Moreover, we found that arginine manipulated the GSH level in a dose-dependent manner. Compensating arginine to the arginine-depleted medium would decrease the GSH (Figure 2B).

These data imply that arginine may modulate ferroptosis sensitivity through manipulating GSH. To test this hypothesis, the GSH synthesis inhibitor was used to reduce GSH in cells with arginine deprivation. L-buthionine-(S,R)-sulfoxinmine (BSO) is a potent inhibitor of γ-glutamylcysteine synthetase, a key enzyme involved in GSH synthesis [26]. Although previous study has reported that BSO exposure would directly initiate ferroptosis [27], we found that the administration of a lower concentration of BSO alone failed to directly initiate ferroptosis in MEF cells with arginine depletion, but indeed restarted erastin-induced cell death in the arginine-depleted medium. In addition, this cell death could be abolished by the ferroptosis inhibitor Ferrostatin-1, suggesting that a lower concentration of BSO could restart erastin-induced ferroptosis during arginine depletion (Figure 2C). Importantly, these lower concentrations of BSO reduced the GSH level in arginine-deprived cells to the level similar to that in cells cultured in the complete medium (Figure 2D). Furthermore, we also compensated the intracellular GSH in cells maintained in the complete medium by the administration of glutathione monoethyl ester (GSHee, a membrane-permeable form of GSH) and found that GSH compensation to the level similar to that in arginine-deprived cells could protect cells from erastin-induced ferroptosis (Figure 2E,F). The lipid peroxidation measurement showed that co-treatment with a lower concentration of BSO would recover erastin-induced lipid peroxidation in cells with arginine depletion (Figure 2G), while compensation of GSH by the administration of GSHee would reduce erastin-induced lipid peroxidation in cells with an arginine presence (Figure 2H). Collectively, these studies suggest that arginine modulates ferroptosis sensitivity through manipulating GSH.

### 2.3. Arginine Deprivation Protects Cells against Ferroptosis through Reducing Fumarate Biosynthesis

To further explore the underlying mechanism by which arginine regulates GSH, the GSH synthesis associated molecules were checked. NRF2 (nuclear factor erythroid 2-related factor 2) is the master transcriptional factor that determines the expression of GSH synthesis associated enzymes and System Xc- subunit SLC7A11 (solute carrier family 7 member 11) [28]. However, arginine depletion failed to change the expression of NRF2 as well as its target genes including SLC7A11 and *Gclc* (glutamate cysteine ligase catalytic subunit), *Gclm* (glutamate cysteine ligase modifier subunit), *Gss* (glutathione synthetase), and *Gsr* (glutathione reductase) (Figure 3A,B). Moreover, arginine deprivation also failed to change the expression of GPX4, the major anti-ferroptotic protein (Figure 3A).

It has been widely reported that some metabolites can directly bind to GSH, resulting in decrease in reactive GSH [29]. Fumarate, a metabolite derived from the arginine-fueled urea cycle (Figure 3C) and several other metabolic pathways, is a reactive α,β-unsaturated electrophilic molecule and can covalently bind to GSH to generate succinicGSH (Figure 3D) [30]. We next wondered whether arginine modulates GSH through sustaining fumarate biosynthesis. Firstly, we observed that arginine deprivation significantly decreased fumarate (Figure 3E). Dimethyl fumarate (DMF), a fumaric acid ester, can easily cross cell membranes, then is immediately hydrolyzed to the active metabolite monomethyl fumarate before finally being converted to fumarate [31,32]. Exposure to a lower concentration of DMF alone failed to directly initiate ferroptosis in MEF cells with arginine depletion, but indeed restarted the erastin-induced ferroptosis, as shown by the PI staining (Figure 3F). More importantly, when DMF was administrated at 10 µM, this recovered ferroptosis in arginine-deprived MEF cells was comparable to that in MEF cells maintained in the complete medium (Figure 3F). Furthermore, 10 µM DMF also reduced the intracellular GSH in cells with arginine deprivation to the level similar to that in cells with an arginine presence (Figure 3G). In addition, we also found that DMF challenge could restore erastin-induced lipid peroxidation in MEF cells with arginine deprivation, as evidenced by the C11-BODIPY 581/591 staining, followed by flow cytometry analysis and confocal microscope imaging (Figure 3H,I). Altogether, these findings suggest that arginine deprivation elevates GSH and protects cells against ferroptosis through reducing fumarate biosynthesis.

### 2.4. Knockdown of Asl Decreases Fumarate and Desensitizes Cells to Erastin-Induced Ferroptosis

As above-mentioned, fumarate is an important metabolite that is derived from several metabolic pathways including urea cycle, purine metabolism, TTP (tyrosine, tryptophan and phenylalanine) metabolism, and tricarboxylic acid cycle [33]. Among them, the urea cycle can be sustained by arginine. In addition, ASL is the key and direct enzyme to generate fumarate by catalyzing argininosuccinate [34]. To test whether ASL is involved in fumarate biosynthesis and ferroptosis regulation, we knocked down *Asl* by the transfection of siRNA. siRNA-*Asl* (si*Asl*-1, si*Asl*-2) could significantly reduce the mRNA level of *Asl*, as evidenced by the RT-qPCR assay (Figure 4A). In this context, we found that knockdown of *Asl* in cells cultured in the complete medium led to a notable reduction in intracellular fumarate, as detected by HPLC (Figure 4B). Importantly, the GSH level was elevated in cells maintained in the complete medium when *Asl* was knocked down (Figure 4C). In addition, the transfection of si*Asl*-1 and si*Asl*-2 effectively inhibited the erastin-induced ferroptosis in the presence of arginine, as shown by the PI staining (Figure 4D). Furthermore, both the flow cytometry analysis and confocal microscope imaging revealed that arginine failed to facilitate erastin-induced lipid ROS when *Asl* was knocked down (Figure 4E,F). These results demonstrate that the knockdown of *Asl* decreases fumarate and desensitizes cells to erastin-induced ferroptosis in the presence of arginine.

### 2.5. Fumarate Is Decreased in Response to Erastin Exposure

How fumarate metabolism manipulates ferroptosis in response to erastin is the next open question. Interestingly, we found that the intracellular fumarate was decreased during erastin exposure in the complete medium, as measured by HPLC (Figure 5A). This indicated that cells initiate a defense approach in response to erastin challenge by decreasing fumarate biosynthesis, and this may be through decelerating the urea cycle, although the detailed mechanism is still unclear.

## 3. Discussion

Ferroptosis, a novel type of programmed cell death defined in the latest decade, is characterized by the crashed antioxidative systems and overwhelming accumulation of lipid peroxides [35]. Increasing evidence has emerged that amino acid metabolism can manipulate the ferroptosis vulnerability in mammalian cells [16]. Specifically, a previous study reported that arginine deprivation could mitigate erastin-induced ferroptosis [24]. Consistent with this study, herein we present compelling evidence to further confirm that arginine deprivation leads to ferroptosis suppression through elevating the intracellular GSH. This is due to the reduced biosynthesis of fumarate, a key metabolite derived from the arginine-fueled urea cycle. Knockdown of *Asl* prevents fumarate biosynthesis, leading to an elevated GSH level and counteracting ferroptosis in the presence of arginine. More importantly, we found that erastin exposure resulted in the reduction in fumarate. This study thus highlights an arginine/fumarate/GSH axis in the modulation of ferroptosis sensitivity (Figure 5B).

Arginine is a non-essential amino acid. Besides sustaining protein translation, arginine can be metabolized into several kinds of downstream bioactive molecules [36]. Nitric oxide (NO) is synthesized from arginine metabolism by nitric oxide synthase (NOS) [37]. NO can rapidly cause the reaction with the superoxide (O_2_^−^) and increase the production of peroxynitrite (ONOO^−^) and peroxynitrous acid (ONOOH), leading to nitroxidative stress and cell death [38,39]. Arginine also functions as a signaling amino acid that supports mTOR activation in the lysosome through SLC38A9 [40], TM4SF5 [41], and CASTOR1 [42] as the lysosomal arginine sensors. Arginine deprivation activates the GCN2/ATF4 pathway and initiates the so-called integrated stress response [43,44]. These studies highlight the critical importance of arginine metabolism in determining cellular physiology and homeostasis. Therefore, it is not unexpected that the depletion of arginine could disrupt these biochemical processes and even lead to cell death. Although arginine can be endogenously synthesized from citrulline and aspartate via the two rate-limiting enzymes, ASS1 and ASL, some types of cancer cells are defective in this de novo synthesis pathway due to the lower expression of these two critical enzymes [45], which makes these cancer cells become arginine auxotroph and highly addictive to exogenous arginine [46]. The targeted depletion of exogenous arginine by utilizing arginine-catabolizing enzymes (including arginase, arginine decarboxylase and arginine deiminase) has been well-recognized as an effective therapy against cancers with arginine auxotroph, especially melanoma [47].

Arginine depletion through arginine-catabolizing enzymes can initiate apoptosis in cancer cells [48,49]. Autophagic cell death, mitochondrial damage, and ROS burst have been reported to engage in this cell death [50,51]. However, some kinds of cancer cells naturally possess higher activity in arginine de novo synthesis. In addition, some other cancer cells may gain acquired resistance to the arginine depletion-based therapy. Mechanistically, the re-expression of ASS1 and/or ASL due to epigenetic modification [49] or transcriptional activation [52] confers this acquired resistance. Therefore, arginine depletion-based therapy would have less efficacy to these cancers. Ferroptosis is gradually regarded to hold great promise for therapeutic design against cancers, especially to those traditional therapy-resistant cancers [53]. Herein, we report that the arginine presence can expedite ferroptotic cell death in MEF and HT1080 cells, indicating that ferroptosis-based cancer therapy might possess better efficacy to those cancers with higher intracellular arginine as well as to those cancers that gained acquired resistance to arginine-catabolizing enzyme-based therapy. Therefore, a further understanding of arginine manipulating ferroptosis vulnerability would help to develop novel therapeutic strategies against cancers.

Conlon and colleagues reported that arginine deprivation suppressing ferroptosis is independent of the inhibition of mTOR activity or the activation of the GCN2/ATF4 pathway [24]. In the current study, we suggest that arginine deprivation inhibits ferroptosis in a dose-dependent manner, which is exactly (at least partially) through manipulating GSH, without affecting GPX4 expression. Arginine presence reduces intracellular GSH. Compensation of GSH to the level similar to that in cells with arginine absence by the administration of GSHee could protect cells with an arginine presence against ferroptosis. Furthermore, a reduction in GSH to the level similar to that in cells with arginine presence by administration of BSO would restart ferroptosis in cells with arginine deprivation. Collectively, these elegant studies suggest that arginine determines ferroptosis sensitivity in a GSH-dependent manner. Furthermore, this GSH-dependent mechanism also explains why arginine deprivation fails to counteract GPX4 inhibition-induced ferroptosis.

It is interesting that a different tendency exists between ROS and lipid ROS during arginine depletion in the control conditions. It is well-known that ROS can be detoxified by GSH [54]. Herein, we suggest that arginine depletion in control conditions leads to the decrease in fumarate and the increase in intracellular GSH, which would trigger ROS scavenging. The accumulation of lipid ROS is orchestrated by distinct pathways. The formation of lipid peroxidation is initiated by oxidants, free radicals, or nonradical species [55]. In a reverse reaction, lipid peroxides are detoxified by the GPX4/GSH axis, FSP1/CoQ10 axis, DHODH/CoQ axis, and GCH1/BH4 axis [56,57]. During the control conditions, arginine presence decreases the intracellular GSH. However, other defense mechanisms (FSP1/CoQ10 axis, DHODH/CoQ axis and GCH1/BH4 axis) may not be affected by arginine in the control conditions. These defense mechanisms could compensate the GPX4/GSH axis and detoxify the basal level of lipid peroxidation. During pro-ferroptotic conditions, the burst of lipid peroxides would exceed the processing capacity of the other defense systems, since the GPX4/GSH axis is the major one. When arginine is depleted, the elevated GSH could sustain the enzymatic activity of GPX4 to detoxify more lipid peroxides, and thus decelerate ferroptosis during erastin exposure.

Arginine depletion elevates the intracellular GSH under both the steady state and pro-ferroptotic condition. It seems that GSH biosynthesis is not involved in this GSH elevation, as arginine deprivation fails to alter the expression of the cystine transporter SLC7A11 subunit and GSH synthesis related enzymes. Moreover, arginine deprivation fails to change the expression of NRF2, the master transcriptional factor determining GSH synthesis. Alternatively, we suggest that arginine depletion elevates GSH by decelerating GSH exhaustion through reducing the generation of fumarate. Compensation of fumarate by the administration of DMF reduces GSH in cells without arginine, leading to the restart of erastin-induced lipid peroxidation and ferroptosis.

Fumarate is a reactive α,β-unsaturated electrophilic metabolite [58]. In this study, we propose that arginine elevates fumarate biosynthesis through the urea cycle, during which ASL hydrolyzes argininosuccinate to generate fumarate. Knockdown of *Asl* significantly decreases fumarate in cells with an arginine presence, leading to counteracting lipid peroxidation and inhibiting ferroptosis.

It has been proposed that fumarate is an oncometabolite due to its electrophilic property [58]. Fumarate accumulation due to the loss-of-function mutation of fumarate hydratase reprograms cellular metabolism and initiates an oncogenic feature [59]. Furthermore, fumarate can modulate several types of programmed cell death including apoptosis [60], pyroptosis [61], and necroptosis [62]. Controversy still exists whether fumarate expedites or suppresses ferroptosis. Some studies have shown that fumarate exposure activates NRF2 to counteract ferroptotic cell death [63,64,65]. However, others have suggested that fumarate accumulation due to fumarate hydratase loss-of-function mutation or direct administration, could directly initiate or expedite ferroptosis in several types of cancer cells [66,67]. Mechanistically, Kerins and colleagues suggested that fumarate facilitates ferroptosis through the non-enzymatic succination of GPX4 [66]. Besides, previous studies have also suggested that fumarate could exhaust GSH by generating succinicGSH [30,68]. In this study, we present solid evidence to confirm that fumarate can facilitate ferroptosis. A decrease in fumarate is important for arginine deprivation to suppress ferroptosis. Similarly, knockdown of *Asl* can reduce fumarate, elevate GSH, and suppress ferroptosis during arginine presence.

It is interesting that erastin exposure leads to a reduction in fumarate, although the metabolic basis is unclear. This suggests that cells initiate a defense approach by decreasing fumarate biosynthesis to decelerate GSH exhaustion in response to pro-ferroptotic insults. In summary, this study expands the understanding of amino acid metabolism in ferroptosis regulation.

## 4. Materials and Methods

### 4.1. Cell Culture

MEF and HT1080 cells were maintained in our laboratory. These cells were cultured in DMEM medium (8122292, Gibco, Billings, MT, USA) supplemented with 10% fetal bovine serum (SH30406, Hyclone, Logan, UT, USA) and 1% penicillin/streptomycin (SV30010, Hyclone), and placed in a humidified incubator with 5% CO_2_ at 37 °C. In the case of arginine deprivation, cells were grown in DMEM/F-12 medium without L-arginine, L-leucine, and L-lysine (D9811-15D, US Biological, Salem, MA, USA) supplemented with 10% fetal bovine serum, 1% penicillin/streptomycin as well as 59.05 mg/L L-leucine (61-90-5, Sigma, St. Louis, MO, USA) and 91.25 mg/L L-lysine (657-27-2, Sigma). In the case of the control group with the complete medium, 59.05 mg/L L-leucine, 91.25 mg/L L-lysine, and 147.5 mg/L L-arginine (1119-34-2, Sigma) were added to the DMEM/F-12 medium without L-arginine, L-leucine, and L-lysine.

### 4.2. Reagents

Erastin (S7242), RSL3 (S8155) and Ferrostatin-1 (S7243) were purchased from Selleck Chemicals (Houston, TX, USA). BSO (83730-53-4) and GSHee (118421-50-4) were purchased from Santa Cruz Biotechnology (Dallas, TX, USA). DMF (624-49-7) was purchased from Sigma-Aldrich. Fumarate (110-17-8) was purchased from TCI America (Seekonk, MA, USA). All reagents were dissolved in DMSO (D4540, Sigma) and stored at −20 °C.

### 4.3. PI Staining

PI staining was used to visualize cell death. For microscopy imaging, the cell medium was discarded after the indicated treatment and the cells were incubated with PBS containing 5 µg/mL PI (25535-16-4, MCE, Romulus, MI, USA). The images were captured by using a fluorescent microscope (Eclipse ti2, Nikon, Tokyo, Japan). For quantitative analysis by flow cytometry, after the indicated treatment, both the floating cells in the medium and the adherent cells were collected by using trypsin digestion and centrifugation. The cell pellets were resuspended with 200 µL PBS containing 5 µg/mL PI and then analyzed by using a flow cytometer (CytoFLEX LX, Beckman Coulter, Brea, CA, USA).

### 4.4. Cell Viability Assay

Cell viability was measured by the CCK-8 method. Briefly, the cells were seeded in 96-well plates at a density of 2 × 10^4^ cells per well. After indicated treatment, the cell medium was discarded, and 100 µL fresh medium containing 10 µL CCK-8 reagent (C0039, Beyotime, Shanghai, China) was added to each well. The plates were incubated in a humidified incubator for 1 h at 37 °C. The absorbance was measured on a microplate reader (Multisakan FC, Thermo Scientific, Waltham, MA, USA) at a wavelength of 450 nm.

### 4.5. Intracellular GSH Content Measurement

After the indicated treatment, the cells were collected by trypsin digestion and washed with PBS. Then, 3 × 10^5^ cells were acquired from each sample and mixed with 60 µL protein detergent S solvent from a GSH measurement kit (S0052, Beyotime). After sufficient vortex, the cells were lysed by three cycles of rapid freeze–thaw using liquid nitrogen and a 37 °C water bath. The samples were rested on ice for 5 min, centrifuged at 10,000× *g* for 10 min at 4 °C, and the supernatants were taken for the following GSH measurement by using a commercial kit (S0052, Beyotime), according to the manufacturer’s instructions.

### 4.6. Lipid Peroxidation and ROS Measurement

For microscopy imaging, cells were seeded in confocal imaging dishes with a glass bottom at a density of 4 × 10^5^ cells per dish. After the indicated treatment, the cells were incubated in fresh medium with the lipid peroxidation fluorescent probe C11-BODIPY 581/591 (5 µM, D3861, Invitrogen, Waltham, MA, USA) or total ROS fluorescent probe CM-H2DCFDA (5 µM, C6827, Invitrogen) at 37 °C for 30 min. Then, the cell medium was discarded and cells were washed twice with PBS. Cells were imaged by a confocal microscope (N-STORM, Nikon). For the flow cytometry analysis, cells were seeded in 24-well plates at a density of 1 × 10^5^ cells per well. After the indicated treatment, the cells were collected by trypsin digestion, washed by PBS, and incubated with PBS containing the 5 µM fluorescent probe C11-BODIPY 581/591 or 5 µM CM-H2DCFDA at 37 °C for 30 min. Cells were analyzed by using the 488-nm laser of a flow cytometer (CytoFLEX LX, Beckman Coulter) for excitation. At least 10,000 single cells were counted from each sample. The fluorescent intensity was quantified by FlowJo v10.

### 4.7. Western Blotting

The cells were lysed by using NP-40 lysis buffer (P0013F, Beyotime) supplemented with 1 mM PMSF (ST2573, Beyotime). The protein concentration was measured by an enhanced BCA protein assay kit (P0009, Beyotime) and then diluted to the same concentration by using NP-40 lysis buffer. An equal amount of protein was separated by SDS-PAGE, and then transferred onto nitrocellulose membranes. The nitrocellulose membranes were incubated with the indicated primary antibody at 4 °C overnight, washed by TBST (TBS buffer supplemented with 0.1% Tween-20) three times, then incubated with the corresponding horseradish peroxidase-conjugated secondary antibody at room temperature for 2 h. After being washed with TBST three times and then by TBS, the immunoblotted bands were detected by using an ultra-high sensitivity ECL kit (GK10008, GLPBIO, Montclair, CA, USA). Protein bands were imaged by a luminescence detection system (BG-gdsAUTO710Pro, Baygene Biotech Company, Beijing, China) and quantified using ImageJ-win64.

The primary antibodies used in this study included anti-GPX4 rabbit pAb (A21440, ABclonal, Woburn, MA, USA), anti-NRF2 mouse mAb (sc-365949, Santa Cruz Biotechnology), anti-SLC7A11 rabbit mAb (A2413, ABclonal), and anti-Actin rabbit mAb (AC026, ABclonal). The HRP-conjugated secondary antibodies used in this study included goat anti-rabbit IgG (SA00001-2, Proteintech, Rosemont, IL, USA) and goat anti-mouse IgG (AS003, ABclonal).

### 4.8. Metabolite Extraction

Cell samples were prepared according to the method reported by D. Jirovsky and W. Wiegrebe [32]. Briefly, the cells in 6-well plates were collected by trypsin digestion and washed with PBS twice, and then the cells were mixed with 150 µL acetonitrile:water (*v*:*v*, 1:1). After sufficient vortex, the cells were lysed by three cycles of rapid freeze–thaw using liquid nitrogen and a 37 °C water bath. The samples were placed on ice for 5 min, centrifuged at 7200× *g* for 20 min at 4 °C, and then the supernatants were collected as the metabolites for the following HPLC measurement.

### 4.9. Fumarate Measurement by HPLC

Fumarate measurement by HPLC (Alliance 2695, Waters, Milford, MA, USA) was performed according to the optimized method reported by D. Jirovsky and W. Wiegrebe [32]. The eluent A (0.1% H_3_PO_4_), B (acetonitrile), C (100% methanol), and D (10% methanol) were degassed by ultrasonic cleaning equipment (GS0203, GRANBO, Shenzhen, China) for 20 min and then placed on the HPLC to remove the air of the mobile phase tubes before sample injection. The HPLC condition for fumarate was performed on a ZORBAX SB-C18 column (5 µm, 4.6 × 250 mm, Agilent, Santa Clara, CA, USA) using 0.1% H_3_PO_4_ and acetonitrile (*v*:*v*, 95:5) as the mobile phases at a flow rate of 1 mL/min for 10 min. Both eluents C and D were utilized to wash and balance the chromatographic column. The column temperature was maintained at 25 °C and the detection was carried out at a wavelength of 214 nm. The injection volume was 50 µL. Finally, the concentration of fumarate was analyzed according to the peak areas shown by the Empower 3 software.

### 4.10. Gene Knockdown with siRNA

The siRNAs were synthesized from GenePharma (Shanghai, China). For siRNA transfection, MEF cells in 6-well plates at 80% density were transfected with 300 pmol siRNAs using 12 µL Lipo8000^TM^ transfection reagents (C0533, Beyotime). The sequences for the *Asl* siRNA were si*Asl*-1: 5′-CAAGUGGCCACUGGAGUCAUCUCUA-3′; and si*Asl*-2: 5′-CCAUCACUCUCAACAGCAUTT-3′. The sequence for the siRNA negative control was siNC: 5′-UUCUCCGAACGUGUCACGUTT-3′.

### 4.11. RT-qPCR

The total RNA from the MEF cells was extracted by using the TRIzol reagent (4992730, Tiangen Biotech, Beijing, China). The RNA was reverse-transcribed into cDNA by using HiScript II Q RT SuperMix (R223, Vazyme, Nanjing, China) according to the manufacturer’s protocol. RT-qPCR was performed by using 2× Universal SYBR Green Fast qPCR Mix (RM21203, ABclonal) on a real-time PCR system (CFX384 Touch, Bio-Rad, Hercules, CA, USA). The relative mRNA expression was normalized to *Actin* and calculated by using the 2^−ΔΔCt^ method. Primer sequences used in this study were listed in Table 1 below.

### 4.12. Statistical Analysis

All data were presented as the mean ± SEM using GraphPad Prism 8 (Version 8.0.1) from at least three independent experiments. The *p* value was determined by the Student’s two-tailed *t* test. *p* < 0.05 was considered as statistical significance.

## Figures and Tables

**Figure 1 ijms-24-14595-f001:**
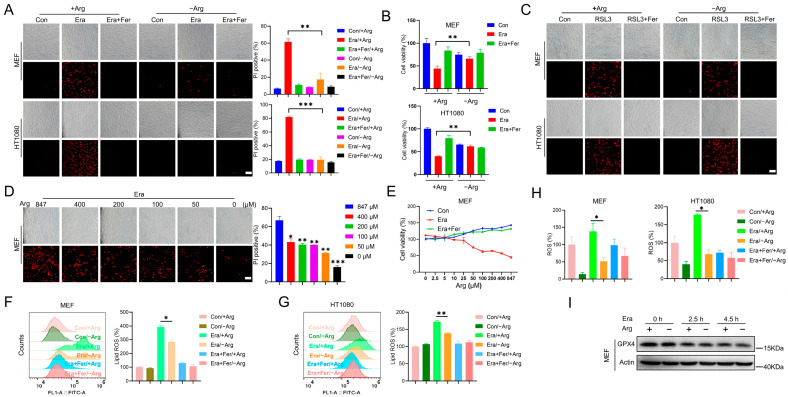
Arginine deprivation inhibits erastin-induced ferroptosis, but not RSL3-induced ferroptosis. (**A**–**C**) MEF and HT1080 cells were cultured in the complete medium for 20 h, and then treated with erastin (2.5 µM) for 6 h and 9 h, respectively or RSL3 (1 µM) for 4 h in the complete medium (with 847 µM arginine) or arginine-depleted medium, with or without Fer (Ferrostatin-1, 10 µM). Cell death was analyzed by PI staining followed by microscopy imaging (**A**,**C**) and flow cytometry analysis (**A**). Scale bar, 100 µm. Cell viability was analyzed by CCK-8 (**B**). (**D**,**E**) MEF cells were treated with erastin (2.5 µM) in the arginine-depleted medium supplemented with the indicated dose of arginine (0, 50, 100, 200, 400, 847 µM) for 6 h. Cell death was analyzed by PI staining followed by microscopy imaging and flow cytometry analysis (**D**). Scale bar, 100 µm. Cell viability was analyzed by CCK-8 (**E**). (**F**–**H**) Lipid ROS (**F**,**G**) and total ROS (**H**) were analyzed by flow cytometry analysis. Representative histograms of flow cytometry and the relative mean fluorescence intensity are shown. (**I**) MEF cells were treated with 2.5 µM erastin (0, 2.5, 4.5 h) in the complete or arginine-depleted medium, and then the GPX4 expression was determined. All data represent the mean ± SEM from three independent experiments. * *p* < 0.05, ** *p* < 0.01, *** *p* < 0.001. +Arg: complete medium, −Arg: arginine-depleted medium, Era: erastin, Fer: Ferrostatin-1.

**Figure 2 ijms-24-14595-f002:**
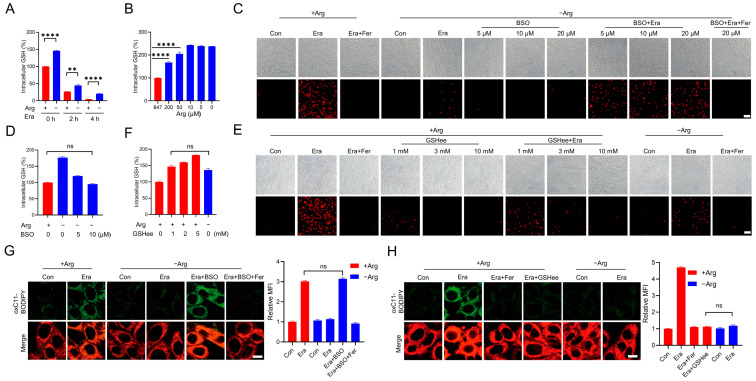
Arginine promotes erastin-induced ferroptosis by reducing GSH. (**A**) Cells were treated with 2.5 µM erastin (0, 2, 4 h) in the complete or arginine-depleted medium. The intracellular GSH was measured. (**B**) Cells were treated with arginine-depleted medium supplemented with the indicated dose of arginine (0, 5, 10, 50, 200, 847 µM) for 6 h. The intracellular GSH was measured. (**C**) Cells were co-treated with BSO (0, 5, 10, 20 µM) and 2.5 µM erastin with or without Fer in the arginine-depleted medium. Cell death was analyzed by PI staining followed by microscopy imaging. Scale bar, 100 µm. (**D**) Cells were treated with BSO (0, 5, 10 µM) in the arginine-depleted medium or left in the complete medium. The intracellular GSH was measured. (**E**) Cells were co-treated with GSHee (0, 1, 3, 10 mM) and 2.5 µM erastin with or without Fer in the complete medium. Cell death was analyzed by PI staining followed by microscopy imaging. Scale bar, 100 µm. (**F**) Cells were treated with GSHee (0, 1, 2, 5 mM) in the complete medium or right in the arginine-depleted medium. The intracellular GSH was measured. (**G**) Cells were treated as in (**C**). Lipid ROS was determined with C11-BODIPY 581/591 staining followed by confocal microscopy imaging. The representative images and relative mean fluorescence intensity are shown. Scale bar, 10 µm. (**H**) Cells were treated as in (**E**). Lipid ROS was determined with C11-BODIPY 581/591 staining followed by confocal microscopy imaging. The representative images and relative mean fluorescence intensity are shown. Scale bar, 10 µm. All data represent the mean ± SEM from three independent experiments. ** *p* < 0.01, **** *p* < 0.0001, ns, not significant. +Arg: complete medium, −Arg: arginine-depleted medium, Era: erastin, Fer: Ferrostatin-1.

**Figure 3 ijms-24-14595-f003:**
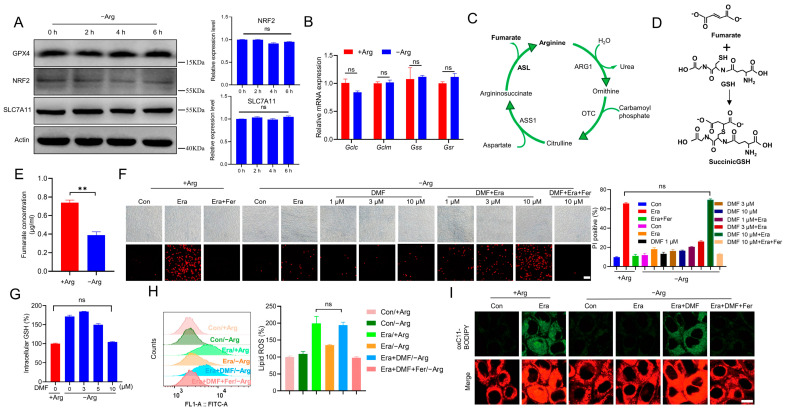
Arginine deprivation protects cells against ferroptosis through reducing fumarate biosynthesis. (**A**) MEF cells were deprived of arginine for the indicated time (0, 2, 4, 6 h), the protein levels of GPX4, NRF2, and SLC7A11 were checked. (**B**) The expression of the NRF2 target genes was determined by RT-qPCR. (**C**) The diagram shows that fumarate is produced through the arginine-fueled urea cycle. ARG1: arginase 1, OTC: ornithine transcarbamylase. (**D**) The diagram shows that fumarate binds to GSH to generate succinicGSH. (**E**) MEF cells were incubated in the arginine-depleted medium for 6 h, and then the fumarate concentration was measured by HPLC. (**F**) Cells were co-treated with DMF (0, 1, 3, 10 µM) and 2.5 µM erastin with or without Fer in the arginine-depleted medium. Cell death was analyzed by PI staining followed by microscopy imaging and flow cytometry analysis. The representative images and the PI positive ratio are shown. Scale bar, 100 µm. (**G**) MEF cells were treated with the indicated concentration of DMF (0, 3, 5, 10 µM) in the arginine-depleted medium, and then the intracellular GSH was measured. (**H**,**I**) Cells were co-treated with DMF (10 µM) and 2.5 µM erastin with or without Fer in the arginine-depleted medium. Lipid ROS was determined with C11-BODIPY 581/591 staining followed by flow cytometry analysis and confocal microscopy imaging. The representative histogram of flow cytometry, relative mean fluorescence intensity, and representative images were shown. Scale bar, 10 µm. All data represent the mean ± SEM from three independent experiments. ** *p* < 0.01, ns, not significant. +Arg: complete medium, −Arg: arginine-depleted medium, Era: erastin, Fer: Ferrostatin-1.

**Figure 4 ijms-24-14595-f004:**
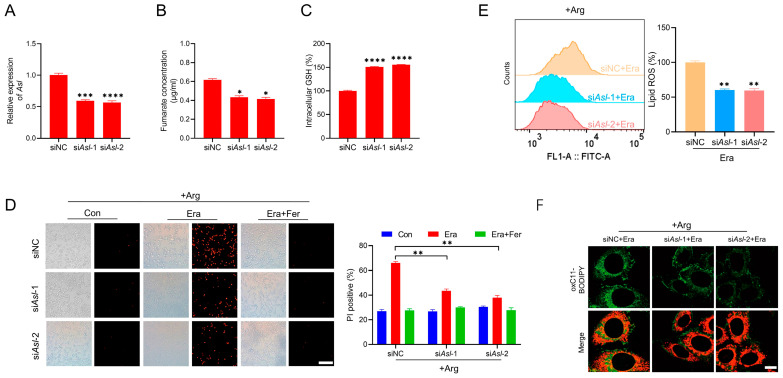
Knockdown of *Asl* decreases fumarate and desensitizes cells to erastin-induced ferroptosis. (**A**) MEF cells were transfected with siRNA-NC or siRNA-*Asl* (si*Asl*-1 and si*Asl*-2), the knockdown efficiency of *Asl* was determined by RT-qPCR. (**B**) Fumarate concentration was measured by HPLC in cells transfected with siRNA-NC or siRNA-*Asl*. (**C**) The GSH level was checked. (**D**) MEF cells were transfected with siRNA-NC or siRNA-*Asl*, then treated with erastin (2.5 µM) for 6 h in the absence or presence of Fer (10 µM) in the complete medium. Cell death was analyzed by PI staining followed by microscopy imaging and flow cytometry analysis. The representative images and the PI positive ratio are shown. Scale bar, 100 µm. (**E**) MEF cells were treated as in (**D**), and lipid peroxidation was measured through C11-BODIPY 581/591 staining followed by flow cytometry analysis. The representative histograms of flow cytometry and the relative mean fluorescence intensity are presented. (**F**) Lipid peroxidation was measured through C11-BODIPY 581/591 staining followed by confocal microscopy imaging and the representative images are shown. Scale bar, 10 µm. All data represent the mean ± SEM from three independent experiments. * *p* < 0.05, ** *p* < 0.01, *** *p* < 0.001, **** *p* < 0.0001. +Arg: complete medium, Era: erastin, Fer: Ferrostatin-1.

**Figure 5 ijms-24-14595-f005:**
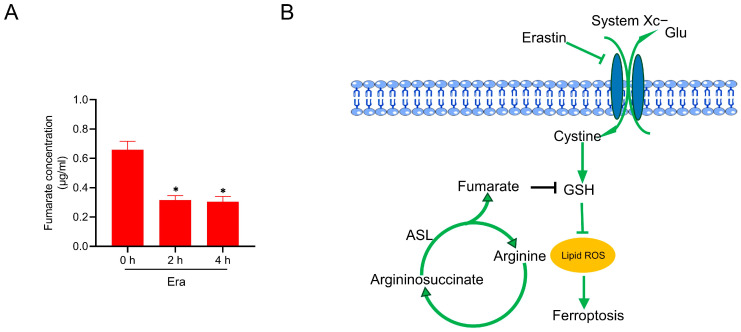
Fumarate is decreased in response to erastin exposure. (**A**) MEF cells were treated with erastin (2.5 µM) in the complete medium. Fumarate was measured by HPLC. (**B**) The summary diagram shows that arginine facilitates erastin-induced ferroptosis through its metabolite fumarate in the urea cycle. Fumarate exhausts GSH by covalently binding to GSH. In response to erastin challenge, fumarate is significantly decreased, suggesting that a protective defense exists to decelerate GSH exhaustion in response to pro-ferroptotic stimuli. All data represent the mean ± SEM from three independent experiments. * *p* < 0.05. Era: erastin.

**Table 1 ijms-24-14595-t001:** Primer sequences used for quantitative real-time polymerase chain reaction.

Gene	Forward Primer (5′-3′)	Reverse Primer (5′-3′)
*Actin*	CATTGCTGACAGGATGCAGAAGG	TGCTGGAAGGTGGACAGTGAGG
*Asl*	GGCAGAGACTAAAGGAGTGGCT	TCGACACTGGATTTCGCTGTGC
*Gclc*	ACACCTGGATGATGCCAACGAG	CCTCCATTGGTCGGAACTCTAC
*Gclm*	TCCTGCTGTGTGATGCCACCAG	GCTTCCTGGAAACTTGCCTCAG
*Gss*	CCAGGAAGTTGCTGTGGTGTAC	GCTGTATGGCAATGTCTGGACAC
*Gsr*	GTTTACCGCTCCACACATCCTG	GCTGAAAGAAGCCATCACTGGTG

## Data Availability

The data presented in this study are available in the article.

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
