# Peer review of "Arginine Expedites Erastin-Induced Ferroptosis through Fumarate"

_ijms, 2023, doi:10.3390/ijms241914595_

Round 1

Reviewer 1 Report

The papers by Guo et al aims at investigating how amino acid metabolism may modulate ferroptosis sensitivity. In particular, they connected the already described effects of arginine depletion on ferroptosis resistance to a decreased buffering effect of fumarate on intracellular GSH. The data well support this thesis and few, additional controls are necessary.

Unfortunately, the Figure are too small, and it is not possible to appreciate the florescence or even the written part of the figures. I strongly recommend the authors to change the images and render them visible.

Can the author specify the time of treatment shown in figure 1? For example, how long did they treat cell with FINs or combination of FINs and Ferrostatin?

What is the standard concentration of Arginine in complete medium? Regarding the graph in Fig 1 E can the author underly the regular concentration of Arginine? 

How many experiments did the author perform to obtain the graph shown in figures? In the legends the authors should include such information.

In figure 1D how did the authors calculate statistical analysis? Why treatment with 200 uM of Arginine has the same p value of 0 uM of Arginine? How many experiments did they include in such analysis? The graph should be representative of 3 independent experiments.

Arginine depletion in control conditions already decreased ROS levels but not Lipid Ros (Fig 1 F and G). The author should discuss this point.   

Did the author test the sensitivity of cells to other Ferroptosis inducers such as FIN56 in combination with Arginine depletion? Did they expect to have the same effect of RSL3?

Fig 2A is not visible. can the author explain how they calculate % of intracellular GSH? How they calculate statistical difference? Did they compare GSH levels always to untreated cells? Since arginine depleted medium per se increases GSH levels, the authors should specify the relative reduction of GSH after exposure to Erastin in control medium and in arginine depleted medium. Figure 2C is not visible and a graph representing the % of PI positive cells should be included.

Is it possible to evaluate SuccinicGSH in their culture system?

Can the authors evaluate the effects (GSH levels, cell death, ROS and lipids peroxidation) of 10 uM of DMF in cells grown in complete medium? Is it already known whether fumarate levels are decreased specifically in ferroptosis resistant cells?

What are the levels of arginine after erastin or other FINs treatment?

Author Response

Dear Reviewer,

Please see the attachment.Thank you and best regards! 

Reviewer 2 Report

The paper by Wu and collaborators reports the effect of arginine on erastin-induced ferroptosis. The results support the author's conclusion that arginine favours ferroptosis by lowering GSH levels. The involvement of fumarate is consistent with reduction of fumarate levels and restoration of erastin-induced ferroptosis by addition of DMF. The origin of fumarate and the full molecular connection with arginine are still unclear, despite the indication that the enzyme ASL is involved. It should be kept in mind that the urea cycle is restricted to hepatic cells and the relevance of this metabolic cycle should be viewed with caution in MEF cells (lines 358-360 and figure 3C). Furthermore, in the discussion the authors should also briefly mention the role of arginine in synthesis of NO and its possible implication in oxidative stress.

Minor points:

1) please define abbreviations in the Introduction (lines 37-38, 42)

2) please increase size of legends in figures and resolution of PI staining images to improve readability. Check and correct typos in figures 3C and 5B.

3) line 129, fluorescent 'radio-probe' is 'ratio-probe'?

moderate editing of English language is required

Author Response

Dear Reviewer,

Please see the attachment. Thank you and best regards!
